# Recent Advances in Research into Jasmonate Biosynthesis and Signaling Pathways in Agricultural Crops and Products

Ruixi Shi, Jinlan Yu, Xiaorong Chang, Liping Qiao, Xia Liu and Laifeng Lu *

Tianjin Key Laboratory of Food Quality and Health, State Key Laboratory of Food Nutrition and Safety, College of Food Science and Engineering, Tianjin University of Science and Technology, Tianjin 300457, China
* Correspondence: frank@tust.edu.cn; Tel.: +86-022-6091-2453

**Abstract:** Jasmonates (JAs) are phospholipid-derived hormones that regulate plant development and responses to environmental stress. The synthesis of JAs and the transduction of their signaling pathways are precisely regulated at multiple levels within and outside the nucleus as a result of a combination of genetic and epigenetic regulation. In this review, we focus on recent advances in the regulation of JA biosynthesis and their signaling pathways. The biosynthesis of JAs was found to be regulated with an autocatalytic amplification mechanism via the MYC2 regulation pathway and inhibited by an autonomous braking mechanism via the MYC2-targeting bHLH1 protein to terminate JA signals in a highly ordered manner. The biological functions of JAs mainly include the promotion of fruit ripening at the initial stage via ethylene-dependent and independent ways, the regulation of mature coloring via regulating the degradation of chlorophyll and the metabolism of anthocyanin, and the improvement of aroma components via the regulation of fatty acid and aldehyde alcohol metabolism in agricultural crops. JA signaling pathways also function in the enhancement of biotic and abiotic stress resistance via the regulation of secondary metabolism and the redox system, and they relieve cold damage to crops through improving the stability of the cell membrane. These recently published findings indicate that JAs are an important class of plant hormones necessary for regulating plant growth and development, ripening, and the resistance to stress in agricultural crops and products.

**Keywords:** jasmonates; auto-regulation; fruit ripening; stress resistance; cold damage



## 1. Introduction

Jasmonates (JAs), including jasmonic acid (JA), jasmonic acid isoleucine (JA-Ile), methyl jasmonate (MeJA), 12-oxo-phytodienoic acid ((9S,13S)-12-oxo-phytodienoic acid, OPDA) and other cyclopentanone derivatives, are an important class of plant hormones necessary for regulating plants' growth and development, resistance to stress, and life cycle [1] (Figure 1). They are widely found in 160 genera and more than 200 species of organisms, including higher plants, mosses, ferns, algae, and fungi, and they are named due to their higher content in the ethereal oil of *Jasminum grandiflorum* flowers. Studies have shown that JAs are abundant in growth sites such as stem ends, young leaves, immature fruits, and root tips [2]. Among them, pH-mediated epimerization and methyl esterification are the main mechanisms for regulating JA activity, and JA-Ile is the main bioactive form of plant endogenous Jas [1]. Active small molecules of Jas in plants also include (+)-7-iso-JA-Leu, (+)-7-iso-JA-Val, (+)-7-iso-JA-Met, and (+)-7-iso-JA-Ala [3]. Propyl dihydrojasmonate (PDJ), a derivative of JA, is currently one of the main commercial forms of JAs catalyzing fruit coloration and shortening fruit ripening.

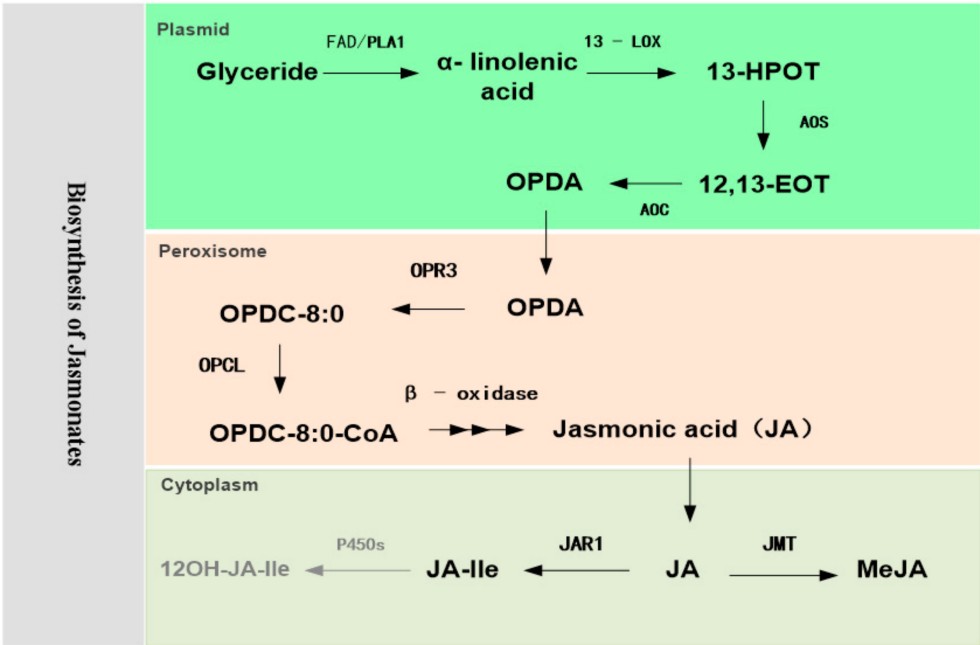

(**a**) Jasmonic acid     (**b**) 12-oxo-phytodienoic acid     (**c**) Methyl jasmonate

(**d**) Jasmonic acid isoleucine     (**e**) Propyl dihydrojasmonate     (**f**) Coronatine

**Figure 1.** The representative structure of molecules of jasmonates (JAs). The JAs include jasmonic acid (C08491), 12-oxo-phytodienoic acid (C13816), methyl jasmonate (C11512), jasmonic acid isoleucine (C18699), propyl dihydrojasmonate (C18538), and coronatine (C16790), which can be retrieved from the website of KEGG (https://www.kegg.jp/, accessed on 25 February 2023).

## 2. Research Progress in Jasmonate Biosynthesis

JA, the chemical name of which is 3-oxo-2-(2′-pentenyl)-cyclopentane acetic acid [3-Oxo-2-(2′-pentenyl)-cyclopentane acid], has a biosynthesis pathway that is based on the unsaturated fatty acid linolenic acid of membrane lipids as the starting point, catalyzed and oxidized by lipid oxygenase [4]. After the action of oxide synthase and cyclase, it is transformed into 12-OPDA, and then reduced and oxidized β-three times into JA, which mainly occurs in chloroplasts and peroxidase bodies. This synthetic pathway is shown in Figure 2.

**Figure 2.** Schematic diagram of the biosynthesis pathway of jasmonates (JAs). Synthesis of JA/JA-Ile from α-linolenic acid generated from glyceride. Abbreviations for compounds: 13-HPOT, (13S)-hydroperoxylinoleic acid; 12,13-EOT, 12,13-epoxyoctadecatrienoicacid; OPDA, 12-oxo-phytodienoic acid; OPDC-8:0, 3-oxo-2(cis-2′-pentenyl)-cyclopentane-1-octanoic acid; PLA1, phospholipase A1; 13-LOX, 13-lipoxygenase; AOS, allene oxide synthase; AOC, allene oxide cyclase; OPR3, OPDA reductase3; OPCL, OPC-8:0 CoA ligase; JAR1, JA–amino acid synthetase; JMT, jasmonic acid carboxyl methyltransferase; CYP94B3, cytochrome P450 oxidases.

Alpha-linolenic acid is the precursor of JA biosynthesis. Glycerides and phospholipids on plant chloroplast membranes contain diene unsaturated fatty acids (18:2), which can be catalyzed by fatty acid desaturase (FAD) to produce triene unsaturated fatty acids (18:3). As expected, silencing *SlFAD7* attenuated JA-Ile accumulation following injury and increased susceptibility to two important insect pests: the chewing herbivores *Spodoptera littoralis* and *Heliothis peltigera* [5]. Interestingly, antisense silencing of *LeFAD7* blocked JA synthesis while increasing the levels of salicylic acid (SA) and pathogenesis-related protein 4 (PR4), reducing the incidence of aphid infestation, whereas JA synthesis mutants, such as *acx1*, *jai1-1*, did not have similar phenotypes [6]. Peach *PpFAD3-1* regulates the synthesis of unsaturated triene fatty acids (18:3), and is also the precursor of volatile short-chain fatty acid flavor substances (E)-2-hexenol and (Z)-3-hexenal [7]. Overexpression of *SlFAD3* and *SlFAD7* resulted in changes in the ratio of (Z)–hex-3-enal/hexanal, resulting in enhanced cold tolerance of tomatoes [8].

Phospholipase A1 (PLA1) hydrolyzes triene unsaturated fatty acids (18:3) in triglycerides and phospholipids to produce free α-linolenic acid (18:3). Plant phospholipases can catalyze the hydrolysis of cell membrane phospholipids, which include PLA, phospholipase C (PLC) and phospholipase D (PLD), depending on the hydrolysis location. PLA cleaves the SN-1 and/or SN-2 positions of glycerophospholipids to release free fatty acids and lysophospholipids. PLA activity, as measured by 14C-lysophosphatidylcholine accumulation in tomato leaves, increased rapidly and systematically in response to wounding, peaking at 15 min and again at 60 min. In addition to JA, systemin, chitosan, and pectin oligosaccharide injury signaling molecules can activate PLA activity in tomato leaves [9].

Free α-linolenic acid (18:3) is oxidized by 13-lipoxygenase (LOX) into 13-hydroperoxylinoleic acid (13-HPOT), which is then dehydrogenated by allene oxide synthase (AOS) to form unstable 12,13-epoxyoctadecatrienoic acid (12,13-EOT). The diversity of oxidized lipids is established by the LOX and enzymes of the unique CYP74 family of the P450 family. First, studies have shown the presence of at least five lipoxygenase genes in tomatoes: *TomloxA*, *TomloxB*, *TomloxC*, *TomloxD*, and *TomloxE*. Mechanical damage, pathogenic bacterial infection, JAs, and system element treatment can activate *TomloxD* expression. Overexpression of *TomloxD* may increase LOX activity and JA content in tomatoes, thereby enhancing tomato resistance to high temperatures, herbivorous insects, and the necrotizing pathogens *Cladosporium fulvum* and *Botrytis cinerea* [10,11]. Suppressor of prosystemin-mediated responses 8 (mutant *spr8*) is a weak resistance plant formed by point mutation of the *TomloxD* catalytic domain. Unlike *TomloxD*, the expression level of *TomloxC* increased with fruit ripening, which mainly regulated the synthesis of flavor substances in C5 and some C6 fruits, independent of tomato stress tolerance [12–14]. Second, CYP74 has three known types: two dehydrogenases, AOS and divinyl ether synthase (DES), and an isomerase, hydroperoxide lyase (HPL). Sites 295 and 297 within the "I-helical central domain" (oxygen binding domain) are the main determinants of CYP74 catalysis in tomatoes [15]. HPL catalyzes the homocleavage isomerization of fatty acid hydroperoxides into short-lived hemiacetals, which play an important role in regulating plant defense and fragrance substance formation. *CsiHPL1* overexpression caused tomatoes to release more constitutive and wound-induced green leaf volatiles (GLVs), including (Z)-hexenal and (Z)-3-hexen-1-ol. Transgenic lines exhibited decreased expression levels of JA-related genes (*SlAOS* and *SlPI-II*) induced by *Prodenia litura* (Fabricius) and *Alternaria alternata* f. sp. *lycopersici* infection and low resistance to the larvae of the tomato-chewing herbivore *P. litura* [16].

Subsequently, 12,13-EOT is rapidly catalyzed by allene oxide cyclase (AOC) to produce (cis)-OPDA. Systemin and JA may activate AOC, while AOC is specifically expressed in tomato vascular bundles. Its post-transcriptional regulatory effects regulate the rapid synthesis of JA in local regions such as wounds and amplify JA signals to regulate local rapid defense responses [17] The results showed that the activity of the tomato AOC gene promoter changes with growth and development, the reproductive process, and the exogenous environment. Before and after flowering, high AOC promoter activity appeared

in sepals and pistils, anthers, and mature pollen. During fruit development, AOC promoter activity was found preferentially in seeds, and activity decreased with maturity in fruit tissue, with the lowest AOC promoter activity in ripening tomatoes. AOC promoter activity was not detectable in fully developed and senescent leaves. Tomato AOC promoter activity was significantly increased after leaves were treated with exogenous environmental stimuli such as wound response signaling compounds (e.g., JA, systemic proteins, or glucose) [2].

OPDA is transported from chloroplasts to peroxisomes via the ABC transporter PXA1 (peroxisomal ABC transporter 1), COMATOSE, or PED3. In the peroxisome, it is reduced to 3-oxo-2 (cis-2'-pentenyl)-cyclopentane-1-octanoic acid (OPC-8:0) by 12-OPDA reductase3 (OPR3). OPR is a small group of flavin-dependent oxidoreductases in plants and derives its name from OPR3 in tomatoes and *Arabidopsis* spp. Silencing tomato *OPR3* expression by RNA interference (RNAi) blocks JA or JA-Ile accumulation after injury, leading to reduced trichome formation and impaired monoterpene and sesquiterpene production, making it more attractive to the specialist herbivore *Manduca sexta* for feeding and oviposition [18]. Notably, the preference for OPR3-RNAi feeding by *M. sexta* is caused by the increased production of cis-3-hexenal in plants, and cis-4-hexenol is a feeding stimulant for hexagonal ladybug larvae. However, compared to the JA-insensitive 1 (*jai1*) mutant, larval development on the surface of *OPR3-RNAi* plants was significantly delayed, demonstrating that OPDA can substitute for JA/JA-Ile in the local induction of defense gene expression to play a role in stress resistance signals [18]. In addition, the JA-deficient 2 mutant (*spr2*) was delayed in embryo development; 35S::SlAOC-RNAi gene silencing plants exhibited reduced seed sets and other phenotypes, while *acx1a* mutant embryos developed normally, implying that OPDA or OPDA-related compounds are involved in regulating proper embryo development [19].

Subsequently, OPC-8:0 is connected to the coenzyme by the coenzyme A ligase (OPC-8:0 CoA, OPCL) to form OPC-8:0-CoA. OPC-8:0-CoA undergoes catalysis by β-oxidase to produce JA. The acyl-CoA oxidase ACX1A is the first step in the β-oxidation phase of tomato JA synthesis, and the *ACX1A* mutation causes tomato plants to lack the local and systemic ability to activate defense protease inhibitors (PIs) that activate the trauma response; therefore, the mutant's ability to resist *M. sexta* is reduced [20]. Grafting experiments have shown that *ACX1A* is necessary for the β-oxidation phase of JA biosynthesis, and the loss of *ACX1A* function does not affect the recognition of this signal in undamaged response leaves [20].

In the cytoplasm, JA is catalyzed by JA-amino acid synthetase to synthesize JA derivatives such as JA-Ile. SlJAR1 regulates rapid biosynthesis of (+)-7-iso-JA-Ile in injured tomato leaves and is the main isomer of JAs accumulation. The JA-Ile accumulation in *SlJAR1-RNAi* tomato was reduced by 50–75% [21]. JA can also be catalyzed by JA carboxyl methyltransferase (JMT) to MeJA [22]. Their recombinant *Camellia sinensis* JMT (CsJMT) protein could transfer methyl groups to JA to produce orchid-like volatile 1R, 2R-MeJA in vitro. Mechanical injury and low-temperature stress could significantly induce the expression of *CsJMT* and the accumulation of MeJA after harvest [23]. JA can also be combined with glycosides and converted into inactive forms for transfer and transport. In addition, cytochrome P450 oxidases (CYP94B3) catalyze the conversion of JA-Ile to inactive or low-active forms such as 12OH-JA-Ile, inhibiting the JA reaction [24].

## 3. Signal Transduction Pathways of the JA Element and Its Feedback Regulation

JAs can activate the signal transduction chain by binding to the nucleus-dispersed coronatine insensitive 1 (COI1) receptor, recruiting the negative regulator JAZ protein of the JA response to degrade it, thereby releasing downstream transcription factors and initiating the JA-regulated plant resistance response, growth, and development (Figure 3). Existing research results show that when the plant is resting, the JA content in the body is low, and a large amount of accumulated JA response inhibitor JAZ protein binds to a variety of co-inhibitors (such as TPL, HDAs, PcG, etc.) and to a series of transcription factors downstream of the JA signal, inhibiting its activity and thereby inhibiting the JA response [4]. When

environmental or developmental factors stimulate JA synthesis in plants, JA content in the body increases rapidly. JA receptor COI1 protein, ASK1 (*Arabidopsis* serine/threonine kinase 1) protein, CUL1 (CULLIN1) protein, and Rbx1 protein are components of the SCF$^{COI1}$ complex. COI1, a protein that plays a vital role in plant growth and development, interacts with and ubiquitinates JAZ proteins, marking them for proteasomal degradation. This process leads to a reduction of the inhibitory effect of JAZ proteins and co-suppressors on transcription factors, ultimately activating JA downstream responses.

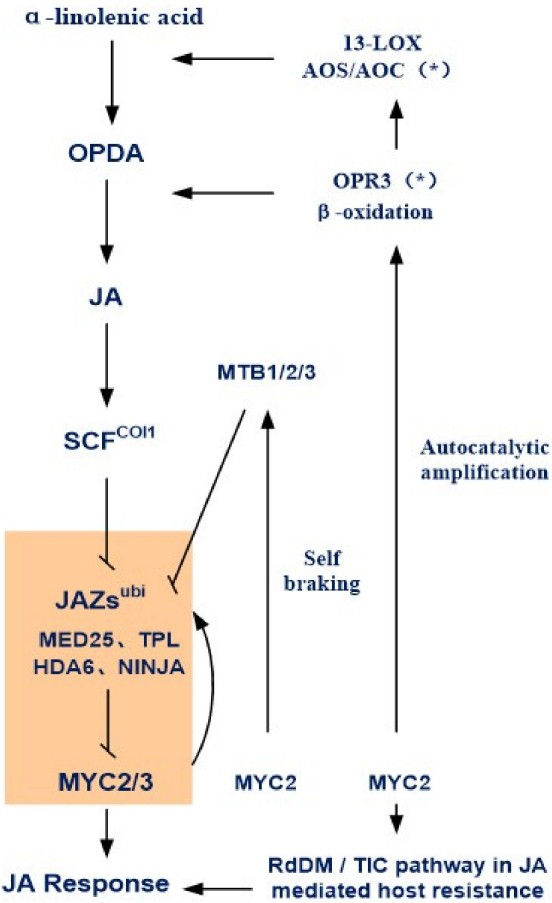

**Figure 3.** Schematic diagram of jasmonate signal induction and the self-regulation mechanism. Jasmonate (JA) activates the signal transduction chain by binding to the coronatine insensitive 1 (COI1) protein, enlisting the jasmonic acid inhibitor JAZ protein for degradation, thereby releasing downstream transcription factors and initiating JA-regulated plant resistance, growth, and development. The JA signal auto-regulates JA synthesis and gene expression, including *LOS*, *AOS*, and *OPR3*, and that of the signaling pathway genes *COI1* and *JMT*. * indicates the key targeted gene in the autocatalytic amplification of JA signaling. Meanwhile, MYC2 activates a small group of JA-inducible basic helix loop helix (bHLH) proteins, termed MYC2-TARGETED BHLH 1 (MTB1), MTB2, and MTB3. It disrupts the formation of the MYC2-MED25 complex and competes with itself to bind to its target gene promoter, inhibiting MYC2 transcriptional activation activity and negatively regulating JA signaling response, which is the so-called 'self-braking' mechanism. 13-LOX, 13-lipoxygenase; AOS, allene oxide synthase; AOC, allene oxide cyclase; OPR3, OPDA reductase 3; OPDA, 12-oxo-phytodienoic acid; JA, jasmonic acid; SCF$^{COI1}$, SCF ubiquitin ligase complex; JAZs, JAs ZIM-domain proteins; NINJA, co-repressor Novel Interactor of JAZ; MED25, subunit 25 of Mediator complex; TPL, co-repressor TOPLESS; HDA6, histone deacetylase 6; MYC2, bHLH Transcription factor; MTB1/2/3, MYC2-Targeted bHLH transcription factor; RdDM, RNA-directed DNA methylation; TIC, TIME FOR COFFEE.

The expression of JA synthesis pathway genes is precisely regulated at multiple levels inside and outside the nucleus, which is a comprehensive result of genetic regulation and epigenetic regulation. The JA signal auto-regulates JA synthesis and gene expression, including *LOS*, *AOS*, and *OPR3*, as well as the signaling pathway genes *COI1* and *JMT*. Strawberry JA was found to positively regulate its own biosynthetic pathway genes, LOS, AOS, and OPR, as well as its associated signaling pathway genes, COI1 and JMT. RNA interference in the grape jasmonic acid pathway gene *VvAOS* in strawberry fruit appeared as fruit un-coloring phenotypes, and the overexpression of the grape JA receptor *VvCOI1* in strawberry fruit accelerated the fruit ripening process [25]. Plant hormones such as ethylene also positively regulate JA synthesis gene expression, which is related to the rapid induction of JA burst. ERFs15 and ERFs16 are transcriptional activators of *TomloxD*, *AOS*, and *OPR3*, key genes in JA biosynthesis. Furthermore, JA-activated MYC2 and ERF16 also function as transcriptional activators of *ERF16*, contributing to dramatic increases in *ERF16* expression. Ethylene-induced *ERF15* and *ERF16* function as potent transcriptional activators that trigger the JA burst in response to herbivore attacks [26]. The epigenetic level of the susceptibility pathogens in *AGO4*-silenced (irAGO4) plants increased, and pathogen infection resulted in lower levels of JA and transcripts of its biosynthetic genes *LOX3* and *OPR3*. Treatment of irAGO4 plants with JA, MeJA, or 12-OPDA restored wild-type resistance [27]. Mechanical damage, insect chewing, pathogen infection, and other external stress conditions will also induce JA synthesis, initiating JA response.

JAs regulate the balance between plant growth and responses to biotic and abiotic stresses. The enhancement of disease resistance in crops or products often inhibits plant growth, development, or the maturation process. To ensure the signal involved in this process is properly managed, an autonomous braking mechanism is needed in order to finely tune JA-mediated stress responses and plant growth (Figure 2). Liu et al. (2019) reported that MYC2 activates a small group of JA-inducible basic helix loop helix (bHLH) proteins, termed MYC2-TARGETED BHLH 1 (MTB1), MTB2, and MTB3; disrupts the formation of its MYC2–MED25 complex and competes with itself to bind to its target gene promoter; inhibits MYC2 transcriptional activation activity; and negatively regulates JA signaling response. MYC2 and MTB proteins form a self-regulating negative feedback pathway to terminate JA signals in a highly ordered manner [28].

## 4. Biological Functions of JAs Signaling Pathway in Agricultural Crops

The main biological functions of jasmonate in plants include regulating the development of stamens and the initiation of epidermal hair; regulating the synthesis of anthocyanins and other secondary metabolism processes and promoting leaf senescence; mediating the resistance of horticultural crops to insects and pathogens; and regulating the response of horticultural crops to stress such as drought, low temperature, and ultraviolet radiation (Figure 4). The specific functions are listed below.

### 4.1. Promotion of the Initial Ripening Process of Fruits

Fruit ripening refers to the process by which fruits achieve the best edible quality through a series of physiological and biochemical changes at the end of growth. Ethylene has been shown to play an important role in fruit ripening. When the ethylene signaling pathway genes *Gr* and *Never ripe* (*Nr*) were mutated, the tomato fruit could not mature [29]. Studies have shown that system 2 ethylene mediated by ACC synthase (ACS1) is regulated by JA induction. JA regulates its signal sensing element MdMYC2 to bind directly to promoters of the ethylene biosynthesis gene *MdACS1* and the ACC oxidase gene *MdACO1* and enhances its transcription to promote ethylene biosynthesis in apple fruit. Interaction with *MdERF3* and the *MdACS1* inhibitor *MdERF2* resulted in increased *MdACS1* transcription [30]. Treatment of 'French pear' (*Pyrus communis* L.) with PDJ can enhance *ACS1* and *ACO1* expression levels to increase ethylene production and promote fruit ripening and can induce an increase in endogenous JA levels [31].

JAs' content changes dynamically with the fruit ripening process, reaching a maximum value before fruit ripening and decreasing gradually. MeJA treatment can promote the development and maturity of strawberry fruit and participate in pigment metabolism, sugar metabolism, fruit softening, and hormone metabolism. The overexpression of *FaAOC* and *FaAOS* can accelerate the ripening of strawberry fruit. The endogenous JA content in strawberry fruit increased sharply from the small fruit stage to the white fruit stage but decreased after fruit maturity, reaching the lowest value at full maturity [32]. JA concentrations were higher in the early growth stages of apple flesh development, decreased with increasing days after flowering, and then increased for pericarp-free ripening. MeJA concentrations in apple flesh initially decreased and then generally increased during the harvest period. The concentrations of JA and MeJA in the pulp of sweet cherry were high during the early growth stages, then decreased towards harvest. PDJ treatment at 104 DAFB (preclimacteric stage) increased the endogenous abscisic acid concentration and anthocyanin concentration at 122 and 131 DAFB (maturation stages) in apples, and relatively high amounts of JA and MeJA are associated with young developing fruit [33].

### 4.2. Chlorophyll and Anthocyanin Metabolism Regulation

Color is an important commodity quality for agricultural crops. Ripening coloration is the result of chlorophyll degradation and the formation or manifestation of carotenoids (yellow or orange) or anthocyanins (purple or red) [34]. It was found that JAs can promote chlorophyll degradation of broccoli (*Brassica oleracea* ssp. *Italica*) and cause postharvest yellowing quality deterioration. MeJA treatment resulted in increased AOC activity, increased endogenous JA levels, strongly reduced maximum quantum yield (Fv/Fm), fluorescence decay rate (Rfd), and total chlorophyll content, promoted phenanthrene oxidase (PAO) activity, and upregulated the expression of chlorophyll degradation genes [35]. MYC2/3/4, an important element of JA signaling, has been shown to directly regulate the chlorophyll metabolism regulatory genes *NON-YELLOW COLORING 1*, *NON-YELLOWING 1*, and *PAO*, triggering Chl breakdown during leaf aging [36]. JA, MeJA, and PDJ treatments enhanced fruit coloration in the early postharvest period of grape berries ("Pione") [37]. In grape suspension culture cells, after addition of JA or light, anthocyan in biosynthesis was enhanced and cell growth was inhibited. The anthocyanins on day 7 after JA (0.02 mM) treatment increased 8.5-fold compared to the control culture [38]. It can be seen that JAs can play an important role in the coloring process of agricultural crops such as fruits by promoting chlorophyll degradation and anthocyanin synthesis.

JAs regulate fruit coloration via ethylene-dependent and independent of the ethylene pathways. Fan et al. (1998) found that exogenous JAs promote ethylene synthesis and fruit coloration in a concentration-dependent manner, and JA treatment stimulates ACC oxidase and ACC synthetase activities. During the onset of ripening in apple and tomato fruits, endogenous JA has a transition peak earlier than ethylene which, together with ethylene, drives the fruit's early ripening process [39]. Moreover, studies have shown that JA might function independently of ethylene to promote lycopene biosynthesis in tomato fruit. The exogenous application of MeJA to *Nr*, the ethylene-insensitive mutant, significantly promoted lycopene accumulation and lycopene biosynthetic gene expression. The lycopene content was significantly reduced in *spr2* and *def1* fruit but increased in 35S::prosys fruits. MeJA treatment significantly enhanced fruit lycopene content and restored lycopene accumulation in mutant *spr2* and *def1* [40].

### 4.3. Regulation of Aroma Components' Metabolism

The combined effect of aroma components can objectively reflect the flavor characteristics and ripeness of agricultural crops such as different fruits. The aroma components of agricultural crops include esters (strawberries, apples, grapes), aldehydes and alcohols (peaches, watermelons, tomatoes), lactones (peaches, coconut, tomatoes), terpenes (nectarines, grapes, oranges), phenols and ethers (bananas, citrus, strawberries), and some sulfur-containing substances. MeJA treatment can increase the content of flavor substances

such as 1-butanol, isoamyl acetate, furfural, 2-methyl butanoic acid, hexanoic acid, octyl butanoate, decyl acetate, jasmolactone, R- and S-furaneol, and dodecylacetatein strawberries [41]. The synthesis of several volatile compounds that are considered positive contributors to grape aroma was improved by MeJA application. MeJA treatment improved p-cymene, methyl jasmonate, and hexanal synthesis, and degraded 2-hexen-1-ol acetate, (Z)-3-hexen-1-ol, and n-hexanol, but the effect of MeJA foliar application on grape volatile content was mainly dependent on vintage [42]. MeJA may be useful in preventing the decline in ester biosynthesis caused by cold storage. MeJA up-regulated the expression of key genes (*PuAAT*, *PuADH3*, *PuADH5*, *PuADH9*, *PuLOX1*, and *PuLOX3*) in the LOX pathway and transcription factors (PuMYB21-like, PuMYB108-like, PuWRKY61, PuWRKY72 and PuWRKY31) in cold storage 'Nanguo' pears, while it increased volatile esters and unsaturated fatty acids [43]. In addition, MeJA treatment increased the concentration of fatty acids and total aromatic volatiles, monoterpenes, sesquiterpenes, aromatic compounds, isoprene, alcohols, and esters in mango pulp and decreased the yield of n-tetradecane [44]. It is worth noting that MeJA treatment significantly promoted the accumulation of volatile organic compounds (VOCs) by inducing the activity of enzymes related to the lipoxygenase pathway and ethylene biosynthesis in tomato fruit, while 1-MCP treatment greatly inhibited the accumulation of VOCs by inhibiting the activity of these enzymes, indicating that MeJA regulated fruit aroma metabolism and the ethylene signaling pathway [45].

Exogenous stimuli such as MeJA affect source–sink changes in aroma biosynthesis and catabolism in tea leaves. Exogenous application of MeJA can induce significant changes in tea volatile components, particularly geraniol, linalool, and its oxides, as well as key aroma precursors such as α-linolenic acid, geranyl diphosphate, farnesyl diphosphate, geranyl diphosphate, and phenylalanine. Moreover, the transcriptome analysis showed increased expression of genes in the α-linolenic acid degradation pathway, which produced massive amounts of JA and quickly activated the holistic JA pathway in tea leaves, while the terpenoid backbone biosynthesis pathway was also significantly affected after MeJA treatment [46,47]. MeJA, as an effective inducer, can induce gene expression in the signal transduction and secondary metabolism of wild chrysanthemum, regulate the biosynthesis pathway of flavor volatiles, and improve aroma quality by promoting flavor volatiles [48].

### 4.4. Participating in the Aging Process

Aging is the final stage of the whole process of development and growth of plants. It is a sort of ordered self-disintegration process regulated by genetic factors. With the initiation of leaf senescence, the degradation and extinction of various organelles (chloroplasts, mitochondria, etc.) in senescent leaf cells proceed in an orderly manner. A large number of senescence-associated genes (SAGs) are expressed in the leaf senescence regulatory network, and SAGs encode proteins that have various physiological functions.

JAs promote senescence in leaves, petals, and other agricultural crop tissues. JA can trigger Chl breakdown during leaf senescence in broccoli [34]. The application of exogenous ethylene and JA can significantly accelerate rose petal senescence by mediating the high expression of the signaling receptor RhMYB108 (R2R3-MYB transcription factor), which regulates *RhNAC053*, *RhNACO92*, and *RhSAG113* genes [49]. MdZAT10 expression was induced by MeJA, which enhanced the transcriptional activity of MdABI5 for *MdNYC1* and *MdNYE1*. MdABI5, an important positive regulator of leaf senescence, significantly accelerated JA-induced senescence in apple leaves. However, a negative feedback pathway also exists for JA-induced leaf senescence, with the JA-responsive protein MdBT2 interacting directly with MdZAT10 and reducing its stability through ubiquitination and degradation, thus delaying MdZAT10-mediated leaf senescence [50].

### 4.5. Regulation of Secondary Metabolism

Secondary metabolism plays an important role in the growth and development of agricultural crops and their resistance to stresses. JAs can induce the synthesis of secondary metabolites such as terpenoids, alkaloids, phenylpropanoids, and flavonoids.

Terpenoids are widely distributed in fruit, flower, leaf, root, and stem tissues and are important components of cell membranes, electron transport carriers, and phytochemicals. In grape cell suspensions, the peak accumulation of sesquiterpenes occurred 72 h after MeJA addition. MeJA-induced sesquiterpenes are synthesized via the MVA pathway, and the induction effect is strictly dependent on cell density in the cell suspension exciton addition [51]. Spraying JAs on lemon leaves significantly increased the content of two phenolic monoterpenoids, namely thymol and carvol [52].

MeJA treatment increased rosmarinic acid synthesis in *Mentha × Piperita* suspension culture cells, which was almost 1.5 times higher than the control sample [53]. In flask and bioreactor cultures, 10 mg $L^{-1}$ JA significantly increased the ginsenoside content. Total ginsenoside and protopanaxadiol ginsenosides increased more than fivefold after 7 days of JA treatment [54]. In marigold-isolated trichome root culture tissues, JA treatment increased both the accumulation of oleanolic acid saponins in hairy root tissue (up to 20-fold) and, in particular, the secretion of these compounds into the medium (up to 113-fold), while inhibiting sterol biosynthesis and accumulation (by about 60%), with some alterations to their profile [55].

JA mediates the R2R3-MYB transcription factor SlMYB14 to participate in the regulation of flavonoid biosynthesis and maintain the dynamic balance of plant active oxygen species. The overexpression of *SlMYB14* in tomatoes leads to increased flavonoid accumulation, and tomato *SlMYB14* acts downstream of *SlMYC2* in the JA signaling pathway [56]. MeJA treatment of *Salvia miltiorrhiza* hairy roots suppressed *SmKFB5* expression while inducing *SmPAL1* and *SmPAL3* transcription and enhanced phenolic acid biosynthesis through transcriptional and post-translational regulatory mechanisms [57]). JA treatment can remove SmJAZ8 inhibition on SmMYB97 and its downstream regulatory genes *PAL1*, *TAT1*, *CPS1*, and *KSL1*. *SmMYB97* overexpression upregulates genes related to these processes and increases phenolic acid and tanshinone biosynthesis [58].

In addition, MeJA treatment of broccoli increased the accumulation of indolyl and aromatic glucosinolates (GSs) in floret tissues [59]. Two sets of WRKY transcription factors (TFs), NnWRKY70a and NnWRKY70b, positively regulated the biosynthesis of BIA in lotus. Both NnWRKY70s were sensitive to JA, and their expression profiles were highly correlated with BIA concentration and BIA pathway gene expression [60].

### 4.6. Regulation of the Redox System to Mediate Stress Resistance

Reduction–oxidation (redox) signaling has been perceived as a balance between low levels of reactive oxygen species (ROS), acting as signals to trigger signaling pathways involved in developmental processes and plant responses to environmental fluctuations, and high levels of ROS, causing oxidative cellular damage. ROS produced in the plant cell can be scavenged or processed by highly efficient antioxidant systems that are composed of antioxidants and antioxidant enzymes [61].

The results showed that MeJA treatment could increase the antioxidant enzymes SOD, POD, CAT, and APX, increase the free-radical-scavenging rate of DPPH, reduce MDA content, and increase the shelf life of strawberries. It can also delay fruit softening and increase sugar content [62]. Foliar spraying with MJ was effective in alleviating the deleterious effects of NaCl-induced salinity stress. Spraying 0.05 and 0.1 mM MeJA on leaves significantly reduced ion leakage and malondialdehyde and hydrogen peroxide production under saline conditions but increased photosynthetic pigments, soluble carbohydrates, proline, total phenols, and flavonoids [63]. MeJA (0.05 mM) treatment up-regulated the expression of antioxidant enzyme genes (*PavPOD*, *PavPPO*, *PavSOD*, and *PavCAT*) and key genes related to JA biosynthesis and signal transduction pathways (*PavLOX*, *PavAOS*, *PavOPR3*, and *PavMYC2*) and enhanced antioxidant enzyme and disease-resistance enzyme activities (catalase, peroxidase, superoxide dismutase, polyphenol oxidase, phenylalanine ammonia-lyase, chitinase, β-1,3-glucanase) to reduce the decay rate of *Alternaria alternata* in post-harvest sweet cherry [64]. JAs are capable of enhancing date palm root resistance to

Foa (*Fusarium oxysporum* f. sp. *albedinis*) infection through the activation of defense-related enzymes such as PPO and POX [65].

In addition, JAs also functions in regulating the antioxidants of fruit. Impaired JA perception (MT-*jai*) led to an increase in the total tocopherol level in mature fruits and enhanced antioxidant capacity [66]. The reason for the contradiction with the antioxidant-promoting effect of JA is related to the seeds. Tocopherols are abundant in tomato seeds, while homozygous MT-*jai* fruits are seedless; the lack of seeds might contribute to the increased levels of tocopherols in the pericarp of these fruits.

### 4.7. The Involvement of JAs in Mediating Crop Disease and Pest Resistance

The JA signal pathway is the main way for plants to resist insect pests. Low concentrations of JAs induced the synthesis of polyphenol oxidase, protease inhibitor, lipoxygenase, and peroxidase in tomato leaves in a dose-dependent manner. Exogenous JA sprayed on farm plants increased the levels of polyphenol oxidase and protease inhibitors, which can stimulate plant resistance against insects in the field [67]. The JA signaling pathway strongly influences midgut protein content in phytophagous insects and supports the hypothesis that amino acids in the insect digestive tract are catabolized by host enzymes and play a role in plant resistance to herbivores [68]. However, the treatment of plants with exogenous applications of JA presumably activates the natural defensive response of plants and thus enhances resistance to attack. For example, treatment of tomatoes with JA at concentrations of 10 mM or 1 mM resulted in fewer but larger fruits, longer ripening times, and delayed fruit setting, indicating that JA-induced defenses caused losses in tomato plant and fruit growth and development [69]. Similar to ethylene, the above findings suggest JAs also have a dual role in defense regulation at the development stage, which resulted in a delay in fruit development, as well as fruit ripening promotion during the ripening stage.

In terms of pathogen infection, JA treatment or the JA signal pathway in strawberry provides the fruit with resistance against gray mold, and JA positively regulates its biosynthetic pathway genes *LOS*, *AOS*, and *OPR* and signaling pathway genes *COI1* and *JMT*. The *SlMYC2* gene plays a key regulatory role in MeJA-induced disease resistance in agricultural crops. Knockout of *SlMYC2* significantly decreased defensive and antioxidant enzyme activities, as well as PR gene expression levels (*SlPR-1* and *SlPR-STH2*) and key genes related to JA biosynthesis and its signaling pathway, which aggravated the disease symptoms [70]. MeJA mediates terpene synthesis of strawberry fruit against *B. cinerea* infection. In strawberry fruit, MeJA induces *FaTPS1* expression through *FaMYC2* and rapidly increases the content of sesquiterpenoids such as daulene D, which improves strawberry resistance against *B. cinerea* infection [71]. The bHLH transcription factor gene *SlJIG*, which participates in JA-induced terpene biosynthesis, is the direct target of MYC2, forms a *MYC2-SlJIG* module, and functions in terpene biosynthesis and resistance against the cotton bollworm and *B. cinerea*. *SlJIG* knockout plants generated by gene editing had a lower terpene content than the wild type due to the lower expression of TERPENE SYNTHASE genes [72].

JAs' regulation of disease resistance involves the epigenetic regulatory mechanism. AGO4-dependent miRNAs play a central role in modulating JA biogenesis and signaling during hemibiotrophic fungal infections. The silence of AGO4 strongly altered the miRNA accumulation dynamics. IrAGO4 plants accumulated lower levels of JAs and fewer transcripts of JA biosynthesis genes. Treating irAGO4 plants with JA, MeJA, or cis-(+)-12-oxo-phytodienoic acid restored the WT susceptibility level [27]. LncRNA4504 may play a vital role in MeJA-induced tomato disease resistance, possibly by promoting the accumulation of total phenols and total flavonoids, enhancing defense enzyme activities, and upregulating JA signal pathway genes' expression. MeJA treatment also promoted transcripts of genes related to JA biosynthesis (*SlLOXD*, *SlAOS*, and *SlAOC*) and its signal transduction (*SlMYC2* and *SlCOI1*), along with increased endogenous JA content. However, lncRNA4504 silencing almost counteracted the effects of MeJA on the above indexes,

leading to higher disease incidence and lesion diameter in the lncRNA4504-silenced+MeJA group than those in the MeJA group [73].

Expression of *LeARG2*, encoding arginase, and arginase activity are important for components of the JA-activated plant defense response. Trauma- and JA-induced *LeARG2* expression was not observed in tomato *jar1* mutants, indicating that this response is strictly dependent on the intact JA signaling pathway [74]. Tomato plants require activation of the JA biosynthetic pathway in response to injury or (pro)-systemin to generate distant signals, and their recognition in distal leaves is dependent on JA signaling, suggesting that JA or a related compound derived from the octadecane pathway may act as a transmissible wound signal [75].

### 4.8. Regulation of Cell Membrane Stability to Relieve Cold Damage

Temperature is an important factor affecting the quality and shelf life of fruits and vegetables in post-harvest storage. Cold damage to fruits and vegetables refers to damage to tissues in cold storage above the freezing point due to internal structural damage to cells and resulting physiological dysfunction, as an adverse response to low-temperature stress. Cold damage leads to surface sinking, wrinkling, fruit-surface browning, flesh browning, and other symptoms in agricultural crops. Ripening bananas, mangoes, peaches, and other fruits lose part of their aroma components with the occurrence of cold damage. In addition, cold damage to fruits and vegetables weakens their ability to resist pathogenic bacteria and increases susceptibility to pathogenic bacteria infestation, leading to rapid decay and odor production, seriously affecting the commercial value [76].

MeJA can alleviate the losses related to cold damage in various fruit varieties by inhibiting the expression of *PpPAL1*, *PpPPO1*, and *PpPOD1/2* and increasing the content of phosphatidylcholine, phosphatidylethanolamine, and phosphatidylglycerol [77]. The JA signaling pathway contributed to the mitigation of chilling injury (CI) in peaches during cold storage. Treatment with 10 M MeJA can delay the decrease in unsaturated fatty acids and increase in saturated fatty acids, improve the unsaturation of fatty acids in the cell membrane, and maintain a higher $\alpha$-linolenic acid concentration and higher LOX, AOS, and AOC activities. The accumulation of JA and JA-Ile enhanced the JA signaling pathway and alleviated the CI index of solute and hard peach cultivars [78]. JA plays an important role in reducing seed browning caused by low temperature in pepper fruits. When treated with 0.05 mM MeJA, seed browning was highly inhibited, and endogenous JA production was increased through the early activation of JA synthesis genes [79].

The mechanism of JAs inhibiting the occurrence of cold damage in peaches currently includes the following two types. First, JA reduces the tolerance of peaches to cold damage and CI based on the regulation of fatty acids in phospholipids. Exogenous MeJA inhibits the expression of *PpPAL1*, *PpPPO1*, *PpPOD1/2*, and *PpFAD8.1*, increasing the expression and JA content of *PpLOX3.1*, *PmMYC2.2*, and *PpCBF3* [77].

MeJA treatment markedly induced *SlMYC2* expression; increased proline content, lycopene content, and antioxidant enzyme activities, including superoxide dismutase, peroxidase, catalase, and ascorbate peroxidase; inhibited electrical conductivity and malondialdehyde content; and effectively reduced CI incidence and the CI index. However, these effects of MeJA treatment were partially counteracted in *SlMYC2*-silenced tomato fruit, and the CI incidence and CI index of *SlMYC2*-silenced fruit increased [80]. Secondly, JAs improve the cold resistance of peach fruits by increasing the amount of ethylene and soluble sugar content. JA treatment reduces the severity of internal browning of peach pulp and does not inhibit fruit softening during the 35-day storage period [81].

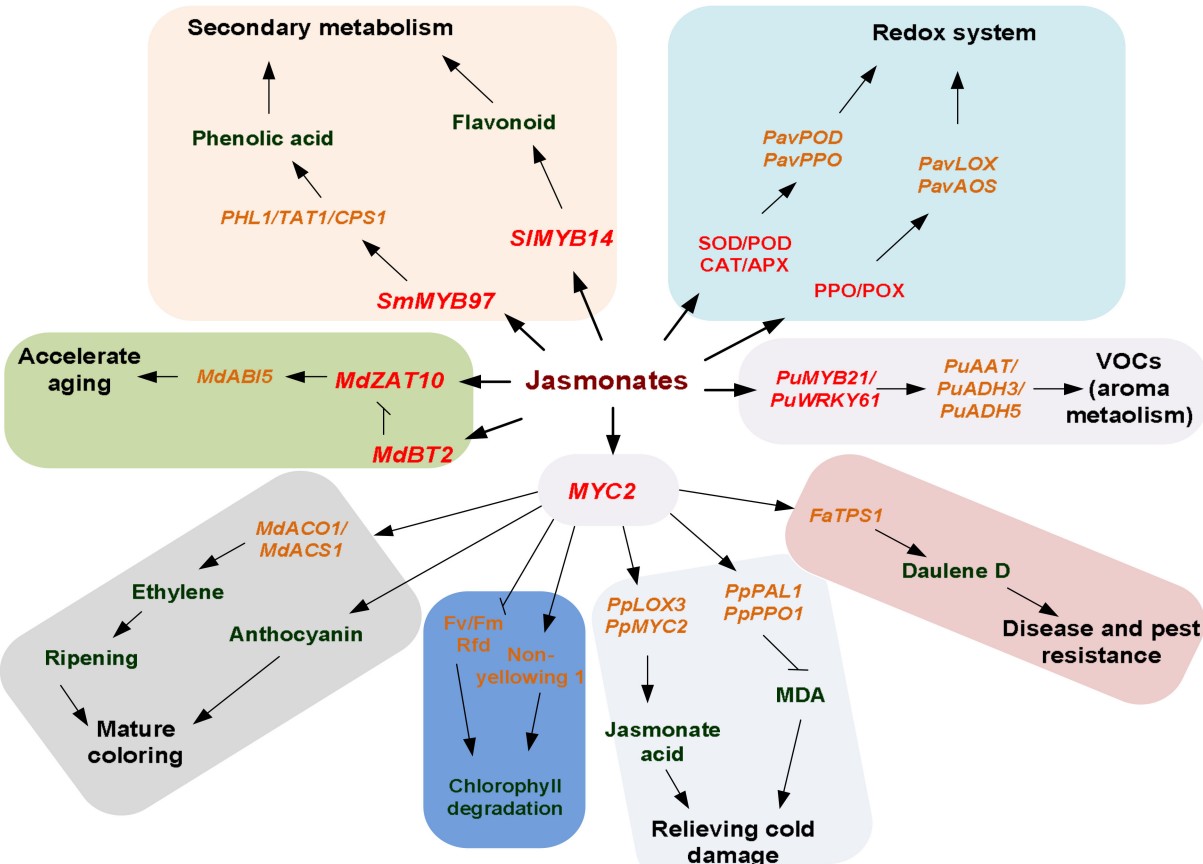

**Figure 4.** Biological functions of the jasmonate signaling pathway in agricultural crops. The biological functions of JAs mainly include the promotion of fruit ripening at the initial stage via ethylene-dependent and independent mechanisms, the regulation of mature coloring via regulating the degradation of chlorophyll and the metabolism of anthocyanin, and the improvement of aroma components via the regulation of fatty acid and aldehyde alcohol metabolism in agricultural crops. Additionally, JA signaling pathways enhanced stress resistance via the regulation of secondary metabolism and the redox system and alleviated cold damage in crops through improving the stability of the cell membrane.

## 5. Conclusions and Future Perspectives

JA synthesis and signaling pathway genes are precisely regulated at multiple levels within and outside the nucleus through a combination of genetic and epigenetic regulation mechanisms. JAs have been shown to auto-regulate the biosynthetic pathway genes *LOS*, *AOS*, and *OPR*, as well as the signaling pathway genes *COI1* and *JMT*. Two ethylene signaling pathways, ERF15 and ERF16, aided in the transcriptional activation of *TomloxD*, *AOS*, and *OPR3*. This activation of these genes ultimately regulated the effect of jasmonates' (JAs') signalization. In addition to JA and ethylene, wounding signaling molecules such as phylon, chitosan, and pectic oligosaccharides, as well as adverse external conditions such as mechanical damage, insect chewing, and pathogenic bacterial infestation, induce JA synthesis in agricultural crops and initiate JA responses. For example, PLA1 phospholipase increases rapidly and systematically after injury. Systemin and JAs can activate propylene cycling enzyme AOC activity, regulate the rapid synthesis of JA in local areas such as wounds, and amplify the JA signal to regulate the local rapid defense response. At the epigenetic level, AGO4 is actively involved in regulating the transcription of genes such as *LOX3* and *OPR3* to mediate plants' responses to pathogenic infections. JAs play a crucial role in the regulation and maturation of agricultural crops. Such effects include the promotion of the ACS1-mediated ethylene system, which regulates the initial maturation

process, the promotion of chlorophyll degradation and anthocyanins to mediate the fruit coloring process, and the metabolism of fatty acids and aldehydes for the synthesis of aroma components. Furthermore, JAs also promote the synthesis of abscisic acid and mediate the aging process of agricultural crops. Additionally, they regulate the secondary metabolism of crops, influence their redox systems, and enhance their resistance to insects, pathogens, and cold damage, thus contributing to their formation and quality maintenance.

More and more studies have shown that JAs, as important plant growth regulators, not only are essential for agricultural crops' defenses against herbivores and response to harsh environmental conditions and other types of biotic and abiotic challenges, but also play an important role in the growth and development of agricultural products, from reproductive development to maturation and senescence. The expression levels and contents of genes related to JA synthesis change between different growth and development stages of crops/agricultural products. For example, the activity of the key AOC gene promoter for JA synthesis changed with growth and development, reproductive processes, and exogenous environmental stimulation, while the transcriptional expression regulation and epigenetic regulation mechanism of JA synthesis genes remained unclear. To ensure the dispersal of seeds by animals, JA signaling desensitization after fruit ripening involves interactions between plant hormones such as JA and abscisic acid, SA, ethylene, etc. Further exploration of the mechanisms underlying these phenomena is needed for the better application of JA substances to improve yield, quality, and nutrition of agricultural crops.

**Author Contributions:** Investigation, R.S., J.Y. and X.C.; writing—original draft preparation, R.S. and L.L.; writing—review and editing, L.L., L.Q. and R.S.; supervision, L.L. and X.L.; project, L.L. and L.Q. All authors have read and agreed to the published version of the manuscript.

**Funding:** This work was funded by the National Natural Science Foundation of China [Grant number 32272395]; Science and Technology Commissioner Foundation of Enterprise of Tianjin [Grant number 22YDTPJC00180]; China Postdoctoral Science Foundation [Grant number 2022M712375], China.

**Conflicts of Interest:** The authors declare no conflict of interest.

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
