# Peer review of "Recent Advances in Research into Jasmonate Biosynthesis and Signaling Pathways in Agricultural Crops and Products"

_processes, doi:10.3390/pr11030736_

Round 1
Reviewer 1 Report
Jasmonates have previously been considered primarily compounds regulating plant stress responses. However, numerous studies have demonstrated that these so-called stress hormones can be involved in various developmental aspects of plant life. Environmental stresses as well as pest attacks seriously affect agricultural yields, and taking into account involvement of JAs in processes such as fruit ripening and coloring, it is highly beneficial to summarize novel approaches and knowledge of function and metabolism of this important phytohormonal group.
However, there are some concerns that need to be resolved. First of all, I’ve got an impression that the content of the manuscript, or we can say the theme of the review, doesn’t quite fit with the theme and scopes of the special issue "Agriculture Products Processing and Storage". Nevertheless, it’s up to Editorial Board to make the final decision.
Starting from the title, authors all through the review emphasize that JAs play distinct important roles in agricultural crops solely, despite the fact that these JA functions are considered and have been shown for plants in general. It also goes for some other biological processes, for example, aging which represent the normal continuation of growth in plants per se. Hence, even though that the data collected and displayed in the review represent meaningful insight in JAs metabolism and function in selected crops, some parts of the text need to be revised as I pointed out in my comments. Speaking of which, I suggest including more crop species, rather than tomato and strawberries for which the majority of data are presented for.
In addition, some of the sentences are obscure and inexplicit. Marked parts need to be properly rephrased with concrete meaning, for clearer interpretation. English language checkup is highly recommended, since there is a need for numerous corrections and addition of missing prepositions.
Finally, certain sections are lacking of appropriate references corresponding to and supporting distinct statements in the text.
The rest of my suggestions and comments are in the PDF file of the manuscript.

Reviewer 2 Report
No comments
Reviewer 3 Report
This manuscript writes well and interests this journal.
Reviewer 4 Report
In this manuscript, the authors provide a comprehensive review of jasmonate regulation, including its biosynthesis and signaling pathways, and its roles in plant development, physiology and biotic and abiotic stress responses.
There are however some problems with this work.
1) English language use is poor and many parts of the manuscript are therefore unclear. The manuscript needs to be better written for it to be properly reviewed.
2) Throughout the manuscript, there are statements made without proper citation (too many to point out here).
3) Incomplete legend for Figure 3: the feedback control mechanisms at the signaling and biosynthesis levels are not covered in the legend.
4) There is often a lack of explanation when it is needed.
For example:
Line 402: “Spraying 0.05 and 0.1 mM MeJA on leaves significantly reduced ion leakage and malondialdehyde and hydrogen peroxide production, but increased photosynthetic pigments, soluble carbohydrates, proline, total phenols, and flavonoids (Rouhollah Karimi et al., 2022).”
This is in contradiction with senescence promotion and requires an explanation. I understand that the scientific literature often has seemingly contradictory findings. But a review should at least attempt to find an explanation for these, or point out that these findings are contradictive.
Line 414: “In addition, impaired JA perception led to an increase in total tocopherol level in 414 mature fruits and enhanced antioxidant capacity (Almeida et al., 2015). “
This is in contradiction with the antioxidants promotion effect of JA and requires an explanation.
Line 425: “However, treatment of tomatoes with JA at concentrations of 10 mM or 1 mM resulted in fewer but larger fruits, longer ripening times, and delayed fruit setting, indicating that JA-induced defenses caused losses in a tomato plant and fruit growth and development (Ahnya et al., 2001).”
This is in contradiction with the fruit ripening effect of JA and requires an explanation.
Line 326: Exogenous stimuli such as MeJA affect source-sink changes in aroma biosynthesis and catabolism in tea leaves.
What does this mean? Not clarified in the subsequent text.
Line 432: “RNAi of the grape JA pathway gene VvAOS in strawberry fruit showed a fruit-nonstaining phenotype; exogenous JA saved this phenotype (Jia et al., 2016).”
What is fruit-nonstaining?
5) There are a number of illogical steps or strange transitions in the text:
For example:
Line 232: “The specific functions of regulating the response of agricultural crops to low temperature, ultraviolet radiation, and stresses are as follows:”
This is then followed by JA effects on development…
Line 266: Fruit ripening coloration is the result of chlorophyll degradation in fruit cells and the formation or manifestation of carotenoids (yellow or orange) or synthetic anthocyanins (purple or red) (Zhang et al., 2007). The study found that JA can promote chlorophyll degradation of broccoli (Brassica oleracea ssp. Italica) and cause postharvest yellowing quality deterioration.
Broccoli is not a fruit.
Also, the anthocyanins are not synthetic.
Line 396: “Agricultural crop cells contain a variety of terminal oxidases, including cytochrome oxidase, alternating oxidase, phenol oxidase, ascorbic acid oxidase, and glycolate oxidase, which make them adapt to various external conditions within a certain range. “
Don’t understand what this sentence is doing here. As the next sentence (see below) does not logically follow.
“The results showed that MeJA treatment could increase antioxidant enzymes SOD, POD, CAT, and APX, increase free radical scavenging rate of DPPH, reduce MDA content, and increase shelf life of strawberries.”
SOD and CAT are not oxidases…
6) There are incorrect statements.
For example:
Line 62: Interestingly, antisense silencing of LeFAD7 blocked JA synthesis while increasing the mRNA levels of salicylic acid (SA)
Salicylic acid doesn’t have mRNA levels.
Line 184: It binds to the active JAs molecule, and then interacts with the JAZ protein substrate and degrades it by ubiquitination.
The receptor does not degrade the JAZ protein. It ubiquitinates it, and it is then degraded by the 26S proteasome.
7) There is unnecessary detail for a review.
For example:
Line 365: “Spraying JA on lemon leaf significantly increased the content of two phenolic monoterpenoids (such as thymol), from 0.42% of the control group to 4.37% (0.40) of JA group and carvol from 0.77% of the control group to 14.76% (0.40) of JA group (Pirbalouti et al., 2019).”
Line 369: “MeJA (0.1 mM) treatment increased rosmarinic acid synthesis in Mentha × Piperita 369 suspension culture cells with a maximum accumulation of 117.95 mg g-1 DW (12% DW).”
The readers can check the exact concentrations and percentages in the research papers.
8) The conclusions and future perspectives section is largely just a repetition of what was already described in the text.
Round 2
Reviewer 1 Report
The authors have improved the manuscript and provided appropriate responses to the previous comments. However, some minor issues quoted in the PDF document of the manuscript need to be additionally addressed.
